# Learning to Reason
# with Third-Order Tensor Products

**Imanol Schlag**
The Swiss AI Lab IDSIA / USI / SUPSI
`imanol@idsia.ch`

**Jürgen Schmidhuber**
The Swiss AI Lab IDSIA / USI / SUPSI
`juergen@idsia.ch`

## Abstract

We combine Recurrent Neural Networks with Tensor Product Representations to learn combinatorial representations of sequential data. This improves symbolic interpretation and systematic generalisation. Our architecture is trained end-to-end through gradient descent on a variety of simple natural language reasoning tasks, significantly outperforming the latest state-of-the-art models in single-task and all-tasks settings. We also augment a subset of the data such that training and test data exhibit large systematic differences and show that our approach generalises better than the previous state-of-the-art.

## 1  Introduction

Certain connectionist architectures based on Recurrent Neural Networks (RNNs) [1–3] such as the Long Short-Term Memory (LSTM) [4, 5] are general computers, e.g., [6]. LSTM-based systems achieved breakthroughs in various speech and Natural Language Processing tasks [7–9]. Unlike humans, however, current RNNs cannot easily extract symbolic rules from experience and apply them to novel instances in a systematic way [10, 11]. They are catastrophically affected by systematic [10, 11] differences between training and test data [12–15].

In particular, standard RNNs have performed poorly at natural language reasoning (NLR) [16] where systematic generalisation (such as rule-like extrapolation) is essential. Consider a network trained on a variety of NLR tasks involving short stories about multiple entities. One task could be about tracking entity locations (*[...] Mary went to the office. [...] Where is Mary?*), another about tracking objects that people are holding (*[...] Daniel picks up the milk. [...] What is Daniel holding?*). If every person is able to perform every task, this will open up a large number of possible person-task pairs. Now suppose that during training we only have stories from a small subset of all possible pairs. More specifically, let us assume *Mary* is never seen picking up or dropping any item. Unlike during training, we want to test on tasks such as *[...] Mary picks up the milk. [...] What is Mary carrying?*. In this case, the training and test data exhibit systematic differences. Nevertheless, a systematic model should be able to infer *milk* because it has adopted a rule-like, entity-independent reasoning pattern that generalises beyond the training distribution. RNNs, however, tend to fail to learn such patterns if the train and test data exhibit such differences.

Here we aim at improving systematic generalisation by learning to deconstruct natural language statements into combinatorial representations [17]. We propose a new architecture based on the Tensor Product Representation (TPR) [18], a general method for embedding symbolic structures in a vector space.

Previous work already showed that TPRs allow for powerful symbolic processing with distributed representations [18], given certain manual assignments of the vector space embedding. However, TPRs have commonly not been trained from data through gradient descent. Here we combine gradient-based RNNs with third-order TPRs to learn combinatorial representations from natural language, training the entire system on NLR tasks via error backpropagation [19–21]. We point

out similarities to systems with Fast Weights [22–24], in particular, end-to-end-differentiable Fast Weight systems [25–27]. In experiments, we achieve state-of-the-art results on the bAbI dataset [16], obtaining better systematic generalisation than other methods. We also analyse the emerging combinatorial and, to some extent, interpretable representations. The code we used to train and evaluate our models is available at *github.com/ischlag/TPR-RNN*.

## 2   Review of the Tensor Product Representation and Notation

The TPR method is a mechanism to create a vector-space embedding of symbolic structures. To illustrate, consider the relation implicit in the short sentences "Kitty the cat" and "Mary the person". In order to store this structure into a TPR of order 2, each sentence has to be decomposed into two components by choosing a so-called filler symbol $f \in \mathcal{F}$ and a role symbol $r \in \mathcal{R}$. Now a possible set of fillers and roles for a unique *role/filler decomposition* could be $\mathcal{F} = \{\text{Kitty}, \text{Mary}\}$ and $\mathcal{R} = \{\text{Cat}, \text{Person}\}$. The two relations are then described by the set of *filler/role bindings*: $\{\text{Kitty/Cat}, \text{Mary/Person}\}$. Let $d, n, j, k$ denote positive integers. A distributed representation is then achieved by encoding each filler symbol f by a filler vector $\mathbf{f}$ in a vector space $V_{\mathcal{F}}$ and each role symbol r by a role vector $\mathbf{r}$ in a vector space $V_{\mathcal{R}}$. In this work, every vector space is over $\mathbb{R}^d, d > 1$. The TPR of the symbolic structures is defined as the tensor $\mathbf{T}$ in a vector space $V_{\mathcal{F}} \otimes V_{\mathcal{R}}$ where $\otimes$ is the tensor product operator. In this example the tensor is of order 2, a matrix, which allows us to write the equation of our example using matrix multiplication:

$$\mathbf{T} = \mathbf{f}_{\text{Kitty}} \otimes \mathbf{r}_{\text{Cat}} + \mathbf{f}_{\text{Mary}} \otimes \mathbf{r}_{\text{Person}} = \mathbf{f}_{\text{Kitty}} \mathbf{r}_{\text{Cat}}^{\top} + \mathbf{f}_{\text{Mary}} \mathbf{r}_{\text{Person}}^{\top} \tag{1}$$

Here, the tensor product — or generalised outer product — acts as a variable *binding* operator. The final TPR representation is a superposition of all bindings via the element-wise addition.

In the TPR method the so-called *unbinding* operator consists of the tensor inner product which is used to exactly reconstruct previously stored variables from $\mathbf{T}$ using an unbinding vector. Recall that the algebraic definition of the dot product of two vectors $\mathbf{f} = (f_1; f_2; ...; f_n)$ and $\mathbf{r} = (r_1; r_2; ...; r_n)$ is defined by the sum of the pairwise products of the elements of $\mathbf{f}$ and $\mathbf{r}$. Equivalently, the tensor inner product $\bullet_{jk}$ can be expressed through the order increasing tensor product followed by the sum of the pairwise products of the elements of the $j$-th and $k$-th order.

$$\mathbf{f} \bullet_{12} \mathbf{r} = \sum_{i=1}^{n} (\mathbf{f} \otimes \mathbf{r})_{ii} = \sum_{i=1}^{n} (\mathbf{f}\mathbf{r}^{\top})_{ii} = \sum_{i=1}^{n} \mathbf{f}_i \mathbf{r}_i = \mathbf{f} \cdot \mathbf{r} \tag{2}$$

Given now the unbinding vector $\mathbf{u}_{\text{Cat}}$, we can then retrieve the stored filler $\mathbf{f}_{\text{Kitty}}$. In the simplest case, if the role vectors are orthonormal, the unbinding vector $\mathbf{u}_{\text{Cat}}$ equals $\mathbf{r}_{\text{Cat}}$. Again, for a TPR of order 2 the unbinding operation can also be expressed using matrix multiplication.

$$\mathbf{T} \bullet_{23} \mathbf{u}_{\text{Cat}} = \mathbf{T}\mathbf{u}_{\text{Cat}} = \mathbf{f}_{\text{Kitty}} \tag{3}$$

Note how the dot product and matrix multiplication are special cases of the tensor inner product. We will later use the tensor inner product $\bullet_{34}$ which can be used with a tensor of order 3 (a cube) and a tensor of order 1 (a vector) such that they result in a tensor of order 2 (a matrix). Other aspects of the TPR method are not essential for this paper. For further details, we refer to Smolensky's work [18, 28, 29].

## 3   The TPR as a Structural Bias for Combinatorial Representations

A drawback of Smolensky's TPR method is that the decomposition of the symbolic structures into structural elements — e.g. f and r in our previous example — are not learned but externally defined. Similarly, the distributed representations $\mathbf{f}$ and $\mathbf{r}$ are assigned manually instead of being learned from data, yielding arguments against the TPR as a connectionist theory of cognition [30].

Here we aim at overcoming these limitations by recognising the TPR as a form of Fast Weight memory which uses multi-layer perceptron (MLP) based neural networks trained end-to-end by stochastic gradient descent. Previous outer product-based Fast Weights [26], which share strong similarities to TPRs of order 2, have shown to be powerful associative memory mechanisms [31, 27]. Inspired by this capability, we use a graph interpretation of the memory where the representations of

a node and an edge allow for the associative retrieval of a neighbouring node. For the context of this work, we refer to the nodes of such a graph as *entities* and to the edges as *relations*. This requires MLPs which deconstruct an input sentence into the source-entity $\mathbf{f}$, the relation $\mathbf{r}$, and the target-entity $\mathbf{t}$ such that $\mathbf{f}$ and $\mathbf{t}$ belong to the vector space $V_{\text{Entity}}$ and $\mathbf{r}$ to $V_{\text{Relation}}$. These representations are then bound together with the binding operator and stored as a TPR of order 3 where we interpret multiple unbindings as a form of graph traversal.

We'll use a simple example to illustrate the idea. For instance, consider the following raw input: "Mary went to the kitchen.". A possible three-way task-specific decomposition could be $\mathbf{f}_{\text{Mary}}$, $\mathbf{r}_{\text{is-at}}$, and $\mathbf{t}_{\text{kitchen}}$. At a later point in time, a question like "Where is Mary?" would have to be decomposed into the vector representations $\mathbf{n}_{\text{Mary}} \in V_{\text{Entity}}$ and $\mathbf{l}_{\text{where-is}} \in V_{\text{Relation}}$. The vectors $\mathbf{n}_{\text{Mary}}$ and $\mathbf{l}_{\text{where-is}}$ have to be similar to the true unbinding vectors $\mathbf{n}_{\text{Mary}} \approx \mathbf{u}_{\text{Mary}}$ and $\mathbf{l}_{\text{where-is}} \approx \mathbf{u}_{\text{is-at}}$ in order to retrieve the previously stored but possibly noisy $\mathbf{t}_{\text{kitchen}}$.

We chose a graph interpretation of the memory due to its generality as it can be found implicitly in the data of many problems. Another important property of a graph inspired neural memory is the combinatorial nature of entities and relations in the sense that any entity can be connected through any relation to any other entity. If the MLPs can disentangle entity-like information from relation-like information, the TPR will provide a simple mechanism to combine them in arbitrary ways. This means that if there is enough data for the network to learn specific entity representations such as $\mathbf{f}_{\text{John}} \in V_{\text{Entity}}$ then it should not require any more data or training to combine $\mathbf{f}_{\text{John}}$ with any of the learned vectors embedded in $V_{\text{Relation}}$ even though such examples have never been covered by the training data. In Section 7 we analyse a trained model and present results which indicate that it indeed seems to learn representations in line with this perspective.

## 4 Proposed Method

RNNs can implement algorithms which map input sequences to output sequences. A traditional RNN uses one or several tensors of order 1 (i.e. a vector usually referred to as the hidden state) to encode the information of the past sequence elements necessary to infer the correct current and future outputs. Our architecture is a non-traditional RNN encoding relevant information from the preceding sequence elements in a TPR $\mathbf{F}$ of order 3.

At discrete time $t, 0 < t \leq T$, in the input sequence of varying length $T$, the previous state $\mathbf{F}_{t-1}$ is updated by the element-wise addition of an update representation $\Delta \mathbf{F}_t$.

$$\mathbf{F}_t \leftarrow \mathbf{F}_{t-1} + \Delta \mathbf{F}_t \tag{4}$$

The proposed architecture is separated into three parts: an input, update, and inference module. The update module produces $\Delta \mathbf{F}_t$ while the inference module uses $\mathbf{F}_t$ as parameters (Fast Weights) to compute the output $\hat{y}_t$ of the model given a question as input. $\mathbf{F}_0$ is the zero tensor.

**Input Module**   Similar to previous work, our model also iterates over a sequence of sentences and uses an input module to learn a sentence representation from a sequence of words [32]. Let the input to the architecture at time $t$ be a sentence of $k$ words with learned embeddings $\{\mathbf{d}_1, ..., \mathbf{d}_k\}$. The sequence is then compressed into a vector representation $\mathbf{s}_t$ by

$$\mathbf{s}_t = \sum_{i=1}^{k} \mathbf{d}_i \odot \mathbf{p}_i, \tag{5}$$

where $\{\mathbf{p}_1, ..., \mathbf{p}_k\}$ are learned position vectors that are equivalent for all input sequences and $\odot$ is the Hadamard product. The vectors $\mathbf{s}$, and $\mathbf{p}$ are in the vector space $V_{\text{Symbol}}$.

**Update Module**   The TPR update $\Delta \mathbf{F}_t$ is defined as the element-wise sum of the tensors produced by a *write*, *move*, and *backlink* function. We abbreviate the respective tensors as $\mathbf{W}$, $\mathbf{M}$, and $\mathbf{B}$ and refer to them as *memory operations*.

$$\Delta \mathbf{F}_t = \mathbf{W}_t + \mathbf{M}_t + \mathbf{B}_t \tag{6}$$

To this end, two entity and three relation representations are computed from the sentence representation $\mathbf{s}_t$ using five separate networks such that

$$\mathbf{e}_t^{(i)} = f_{\mathbf{e}^{(i)}}(\mathbf{s}_t; \theta_{\mathbf{e}^{(i)}}), 1 \leq i < 3 \tag{7}$$

$$\mathbf{r}_t^{(j)} = f_{\mathbf{r}^{(j)}}(\mathbf{s}_t; \theta_{\mathbf{r}^{(j)}}), 1 \leq j < 4 \tag{8}$$

where $f$ is an MLP network and $\theta$ its weights.

The write operation allows for the storage of a new node-edge-node association $(\mathbf{e}_t^{(1)}, \mathbf{r}_t^{(1)}, \mathbf{e}_t^{(2)})$ using the tensor product where $\mathbf{e}_t^{(1)}$ represents the source entity, $\mathbf{e}_t^{(2)}$ represents the target entity, and $\mathbf{r}_t^{(1)}$ the relation connecting them. To avoid superimposing the new association onto a possibly already existing association $(\mathbf{e}_t^{(1)}, \mathbf{r}_t^{(1)}, \hat{\mathbf{w}}_t)$, the previous target entity $\hat{\mathbf{w}}_t$ has to be retrieved and subtracted from the TPR. If no such association exists, then $\hat{\mathbf{w}}_t$ will ideally be the zero vector.

$$\hat{\mathbf{w}}_t = (\mathbf{F}_t \bullet_{34} \mathbf{e}_t^{(1)}) \bullet_{23} \mathbf{r}_t^{(1)} \tag{9}$$

$$\mathbf{W}_t = -(\mathbf{e}_t^{(1)} \otimes \mathbf{r}_t^{(1)} \otimes \hat{\mathbf{w}}_t) + (\mathbf{e}_t^{(1)} \otimes \mathbf{r}_t^{(1)} \otimes \mathbf{e}_t^{(2)}) \tag{10}$$

While the write operation removes the previous target entity representation $\hat{\mathbf{w}}_t$, the move operation allows to rewrite $\hat{\mathbf{w}}_t$ back into the TPR with a different relation $\mathbf{r}_t^{(2)}$. Similar to the write operation, we have to retrieve and remove the previous target entity $\hat{\mathbf{m}}_t$ that would otherwise interfere.

$$\hat{\mathbf{m}}_t = (\mathbf{F}_t \bullet_{34} \mathbf{e}_t^{(1)}) \bullet_{23} \mathbf{r}_t^{(2)} \tag{11}$$

$$\mathbf{M}_t = -(\mathbf{e}_t^{(1)} \otimes \mathbf{r}_t^{(2)} \otimes \hat{\mathbf{m}}_t) + (\mathbf{e}_t^{(1)} \otimes \mathbf{r}_t^{(2)} \otimes \hat{\mathbf{w}}_t) \tag{12}$$

The final operation is the backlink. It switches source and target entities and connects them with yet another relation $\mathbf{r}_t^{(3)}$. This allows for the associative retrieval of the neighbouring entity starting from

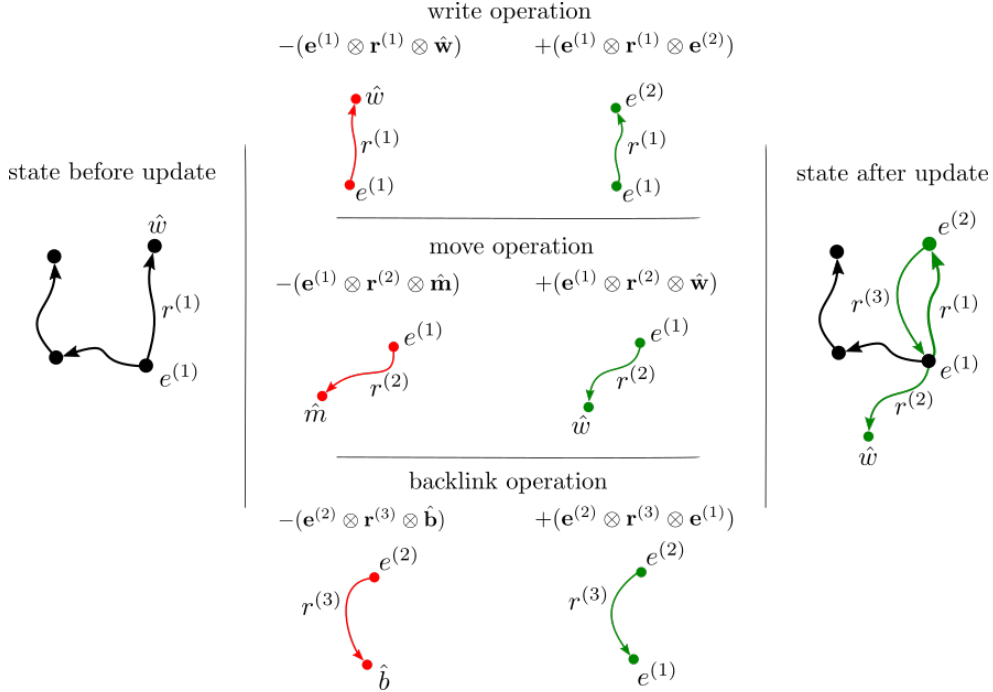

Figure 1: Illustration of our memory operations for a single time-step given some previous state. Each arrow is represented by a tensor of order 3. The superposition of multiple tensors defines the current graph. Red arrows are subtracted from the state while green arrows are added. In this illustration, $\hat{w}$ exists but $\hat{m}$ and $\hat{b}$ do not yet — they are zero vectors. Hence, the two constructed third-order tensors that are subtracted according to the move and backlink operation will both be zero tensors as well. Note that the associations are not necessarily as discrete as illustrated. Best viewed in color.

either one but with different relations (e.g. *John is left of Mary* and *Mary is right of John*).

$$\hat{\mathbf{b}}_t = (\mathbf{F}_t \bullet_{34} \mathbf{e}_t^{(2)}) \bullet_{23} \mathbf{r}_t^{(3)} \tag{13}$$

$$\mathbf{B}_t = -(\mathbf{e}_t^{(2)} \otimes \mathbf{r}_t^{(3)} \otimes \hat{\mathbf{b}}_t) + (\mathbf{e}_t^{(2)} \otimes \mathbf{r}_t^{(3)} \otimes \mathbf{e}_t^{(1)}) \tag{14}$$

**Inference Module**   One of our experiments requires a single prediction after the last element of an observed sequence (i.e. the last sentence). This final element is the question sentence representation $\mathbf{s}_Q$. Since the inference module does not edit the TPR memory, it is sufficient to compute the prediction only when necessary. Hence we drop index $t$ in the following equations. Similar to the update module, first an entity $\mathbf{n}$ and a set of relations $\mathbf{l}_j$ are extracted from the current sentence using four different networks.

$$\mathbf{n} = f_{\mathbf{n}}(\mathbf{s}_Q; \theta_{\mathbf{n}}) \tag{15}$$

$$\mathbf{l}_j = f_{\mathbf{l}_j}(\mathbf{s}_Q; \theta_{\mathbf{l}_j}), 1 \leq j < 4 \tag{16}$$

The extracted representations are used to retrieve one or several previously stored associations by providing the necessary unbinding vectors. The values of the TPR can be thought of as context-specific weights which are not trained by gradient descent but constructed incrementally during inference. They define a function that takes the entity $\mathbf{n}$ and relations $\mathbf{l}_i$ as an input. A simple illustration of this process is shown in Figure 2.

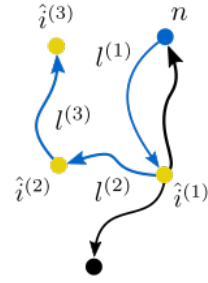

The most basic retrieval requires one source entity $\mathbf{n}$ and one relation $\mathbf{l}_1$ to extract the first target entity. We refer to this retrieval as a one-step inference $\hat{\mathbf{i}}^{(1)}$ and use the additional extracted relations to compute multi-step inferences. Here LN refers to layer normalization [33] which includes a learned scaling and shifting scalar. As in other Fast Weight work, LN improves our training procedure which is possibly due to making the optimization landscape smoother [34].

Figure 2: Illustration of the inference procedure. Given an entity and three relations (blue) we can extract the inferred representations $\hat{i}^{(1:3)}$ (yellow).

$$\hat{\mathbf{i}}^{(1)} = \mathsf{LN}((\mathbf{F}_t \bullet_{34} \mathbf{n}) \bullet_{23} \mathbf{l}^{(1)}) \tag{17}$$

$$\hat{\mathbf{i}}^{(2)} = \mathsf{LN}((\mathbf{F}_t \bullet_{34} \hat{\mathbf{i}}^{(1)}) \bullet_{23} \mathbf{l}^{(2)}) \tag{18}$$

$$\hat{\mathbf{i}}^{(3)} = \mathsf{LN}((\mathbf{F}_t \bullet_{34} \hat{\mathbf{i}}^{(2)}) \bullet_{23} \mathbf{l}^{(3)}) \tag{19}$$

Finally, the output $\hat{\mathbf{y}}$ of our architecture consists of the sum of the three previous inference steps followed by a linear projection $Z$ into the symbol space $V_{\text{Symbols}}$ where a softmax transforms the activations into a probability distribution over all words from the vocabulary of the current task.

$$\hat{\mathbf{y}} = \mathsf{softmax}(Z \sum_{i=1}^{3} \hat{\mathbf{i}}^{(i)}) \tag{20}$$

## 5   Related Work

To our knowledge, the proposed method is the first Fast Weight architecture with a TPR or tensor of order 3 trained on raw data by backpropagation [19–21]. It is inspired by an earlier, adaptive, backpropagation-trained, end-to-end-differentiable, outer product-based, Fast Weight RNN architecture with a tensor of order 2 (1993) [26]. The latter in turn was partially inspired by previous ideas, most notably Hebbian learning [35]. Variations of such outer product-based Fast Weights were able to generalise in a variety of small but complex sequence problems where standard RNNs tend to perform poorly [31, 27, 36]. Compare also early work on differentiable control of Fast Weights [37].

Previous work also utilised TPRs of order 2 for simpler associations in the context of image-caption generation [38], question-answering [39], and general NLP [40] challenges with a gradient-based optimizer similar to ours.

Given the sequence of sentences of one sample, our method produces a final tensor of order 3 that represents the current task-relevant state of the story. Unfolded across time, the MLP representations

can to some extent be related to components of a canonical polyadic decomposition (CPD, [41]). Over recent years, CPD and various other tensor decomposition methods have shown to be a powerful tool for a variety of Machine Learning problems [42]. Consider, e.g., recent work which applies the tensor-train decomposition to RNNs [43, 44].

RNNs are popular choices for modelling natural language. Despite ongoing research in RNN architectures, the good old LSTM [4] has been shown to outperform more recent variants [45] on standard language modelling datasets. However, such networks do not perform well in NLR tasks such as question answering [16]. Recent progress came through the addition of memory and attention components to RNNs. For the context of question answering, a popular line of research are memory networks [46–50]. But it remains unclear whether mistakes in trained models arise from imperfect logical reasoning, knowledge representation, or insufficient data due to the difficulty of interpreting their internal representations [51].

Some early memory-augmented RNNs focused primarily on improving the ratio of the number of trainable parameters to memory size [26]. The Neural Turing Machine [52] was among the first models with an attention mechanism over external memory that outperformed standard LSTM on tasks such as copying and sorting. The Differentiable Neural Computer (DNC) further refined this approach [53, 54], yielding strong performance also on question-answering problems.

# 6  Experiments

We evaluate our architecture on bAbI tasks, a set of 20 different synthetic question-answering tasks designed to evaluate NLR systems such as intelligent dialogue agents [16]. Every task addresses a different form of reasoning. It consists of the story - a sequence of sentences - followed by a question sentence with a single word answer. We used the train/validation/test split as it was introduced in v1.2 for the 10k samples version of the dataset. We ignored the provided supporting facts that simplify the problem by pointing out sentences relevant to the question. We only show story sentences *once* and *before* the query sentence, with no additional supervision signal apart from the prediction error.

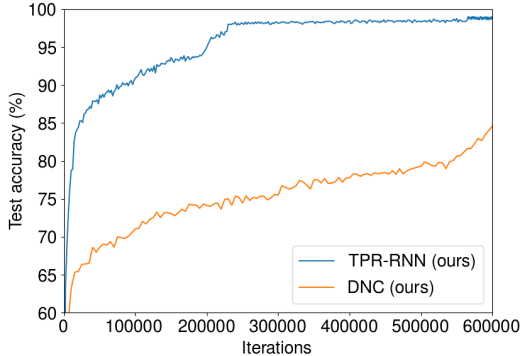

Figure 3: Training accuracy on all bAbI tasks over the first 600k iterations. All our all-tasks models achieve <5% error in 48 hours (i.e. 250k steps). We stopped training our own implementation of the DNC [53] after roughly 7 days (600k steps) and instead compare accuracy in Table 1 using previously published results.

We experiment with two models. The *single-task* model is only trained and tested on the data from one task but uses the same computational graph and hyper-parameters for all. The *all-tasks* model is a scaled up version trained and tested on all tasks simultaneously, using only the default hyper-parameters. More details such as specific hyper-parameters can be found in Appendix A.

In Table 1 and 2 we compare our model to various state-of-the-art models in the literature. We added best results for a better comparison to earlier work which did not provide statistics generated from multiple runs. Our system outperforms the state-of-the-art in both settings. We also seem to outperform the DNC in convergence speed as shown in Figure 3.

Table 1: Mean and variance of the test error for the all-task setting. We perform early stopping according to the validation set. Our statistics are generated from 10 runs.

| Task | REN [55] | DNC [53] | SDNC [54] | TPR-RNN (ours) |
|---|---|---|---|---|
| Avg Error | $9.7 \pm 2.6$ | $12.8 \pm 4.7$ | $6.4 \pm 2.5$ | $\mathbf{1.34} \pm 0.52$ |
| Failure (>5%) | $5 \pm 1.2$ | $8.2 \pm 2.5$ | $4.1 \pm 1.6$ | $\mathbf{0.86} \pm 1.11$ |

Table 2: Mean and variance of the test error for the single-task setting. We perform early stopping according to the validation set. Statistics are generated from 5 runs. We added best results for comparison with previous work. Note that only our results for task 19 are unstable where different seeds either converge with perfect accuracy or fall into a local minimum. It is not clear how much previous work is affected by such issues.

| Task | LSTM [48] best | N2N [48] best | DMN+ [50] best | REN [55] best | TPR-RNN (ours) best | mean |
|---|---|---|---|---|---|---|
| 1 | 0.0 | 0.0 | 0.0 | 0.0 | 0.0 | $0.02 \pm 0.05$ |
| 2 | 81.9 | 0.3 | 0.3 | 0.1 | 0.0 | $0.06 \pm 0.09$ |
| 3 | 83.1 | 2.1 | 1.1 | 4.1 | 1.2 | $1.78 \pm 0.58$ |
| 4 | 0.2 | 0.0 | 0.0 | 0.0 | 0.0 | $0.02 \pm 0.04$ |
| 5 | 1.2 | 0.8 | 0.5 | 0.3 | 0.5 | $0.61 \pm 0.17$ |
| 6 | 51.8 | 0.1 | 0.0 | 0.2 | 0.0 | $0.22 \pm 0.19$ |
| 7 | 24.9 | 2.0 | 2.4 | 0.0 | 0.5 | $2.78 \pm 1.81$ |
| 8 | 34.1 | 0.9 | 0.0 | 0.5 | 0.1 | $0.47 \pm 0.45$ |
| 9 | 20.2 | 0.3 | 0.0 | 0.1 | 0.0 | $0.14 \pm 0.13$ |
| 10 | 30.1 | 0.0 | 0.0 | 0.6 | 0.3 | $1.24 \pm 1.30$ |
| 11 | 10.3 | 0.0 | 0.0 | 0.3 | 0.0 | $0.14 \pm 0.11$ |
| 12 | 23.4 | 0.0 | 0.0 | 0.0 | 0.0 | $0.04 \pm 0.05$ |
| 13 | 6.1 | 0.0 | 0.0 | 1.3 | 0.3 | $0.42 \pm 0.11$ |
| 14 | 81.0 | 0.2 | 0.2 | 0.0 | 0.0 | $0.24 \pm 0.29$ |
| 15 | 78.7 | 0.0 | 0.0 | 0.0 | 0.0 | $0.0 \pm 0.0$ |
| 16 | 51.9 | 51.8 | 45.3 | 0.2 | 0.0 | $0.02 \pm 0.045$ |
| 17 | 50.1 | 18.6 | 4.2 | 0.5 | 0.4 | $0.9 \pm 0.69$ |
| 18 | 6.8 | 5.3 | 2.1 | 0.3 | 0.1 | $0.64 \pm 0.33$ |
| 19 | 31.9 | 2.3 | 0.0 | 2.3 | 0.0 | $12.64 \pm 17.39$ |
| 20 | 0.0 | 0.0 | 0.0 | 0.0 | 0.0 | $0.0 \pm 0.00$ |
| Avg Error | 36.4 | 4.2 | 2.8 | 0.5 | **0.17** | $1.12 \pm 1.19$ |
| Failure (>5%) | 16 | 3 | 1 | **0** | **0** | $0.4 \pm 0.55$ |

Table 3: Summary results of the ablation experiments. We experimented with 3 variations of memory operations in order to analyse their necessity with regards to single-task performance. The results indicate that the move operation is in general less important than the backlink operation.

| Operations | Failed tasks (err > 5%) |
|---|---|
| **W** | 3, 6, 9, 10, 12, 13, 17, 19 |
| **W + M** | 9, 10, 13, 17 |
| **W + B** | 3 |

**Ablation Study** We ran ablation experiments on every task to assess the necessity of the three memory operations. The experimental results in Table 3 indicate that a majority of the tasks can be solved by the write operation alone. This is surprising at first because for some of those tasks the symbolic operations that a person might think of as ideal typically require more complex steps than what the write operation allows for. However, the optimizer seems to be able to find representations that overcome the limitations of the architecture. That said, more complex tasks do benefit from the additional operations without affecting the performance on simpler tasks.

## 7   Analysis

Here we analyse the representations produced by the MLPs of the update module. We collect the set of unique sentences across all stories from the validation set of a task and compute their respective entity and relation representations $\mathbf{e}^{(1)}$, $\mathbf{e}^{(2)}$, $\mathbf{r}^{(1)}$, $\mathbf{r}^{(2)}$, and $\mathbf{r}^{(3)}$. For each representation we then hierarchically cluster all sentences based on their cosine similarity. In Figure 4 we show such similarity matrices for a model trained on task 3. The image based on $\mathbf{e}^{(1)}$ shows 4 distinct clusters which indicate that learned representations are almost perfectly orthogonal. By comparing the

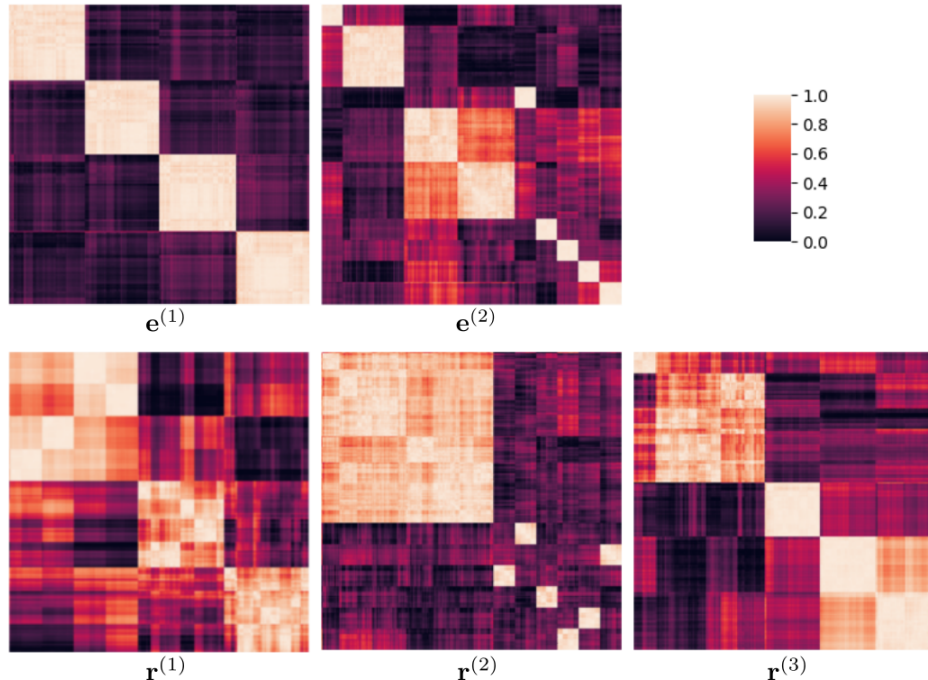

Figure 4: The hierarchically clustered similarity matrices of all unique sentences of the validation set of task 3. We compute one similarity matrix for each representation produced by the update module using the cosine similarity measure for clustering.

sentences from different clusters it becomes apparent that they represent the four entities independent of other factors. Note that the dimensionality of this vector space is 15 which seems larger than necessary for this task.

In the case of $\mathbf{r}^{(1)}$ we observe that sentences seem to group into three, albeit less distinct, clusters. In this task, the structure in the data implies three important events for any entity: moving to any location, bind with any object, and unbind from a previously bound object; all three represented by a variety of possible words and phrases. By comparing sentences from different clusters, we can clearly associate them with the three general types of events.

We observed clusters of similar discreteness in all tasks; often with a semantic meaning that becomes apparent when we compare sentences that belong to different clusters. We also noticed that even though there are often clean clusters they are not always perfectly combinatorial, e.g., in $\mathbf{e}^{(2)}$ as seen in Figure 4, we found two very orthogonal clusters for the target entity symbols $\{\mathtt{t}_{Kitchen}, \mathtt{t}_{Bathroom}\}$ and $\{\mathtt{t}_{Garden}, \mathtt{t}_{Hallway}\}$.

**Systematic Generalisation** We conduct an additional experiment to empirically analyse the model's capability to generalise in a systematic way [10, 11]. For this purpose, we join together all tasks which use the same four entity names with at least one entity appearing in the question (i.e. tasks 1, 6, 7, 8, 9, 11, 12, 13). We then augment this data with five new entities such that the train and test data exhibit systematic differences. The stories for a new entity are generated by randomly sampling 500 story/question pairs from a task such that in 20% of the generated

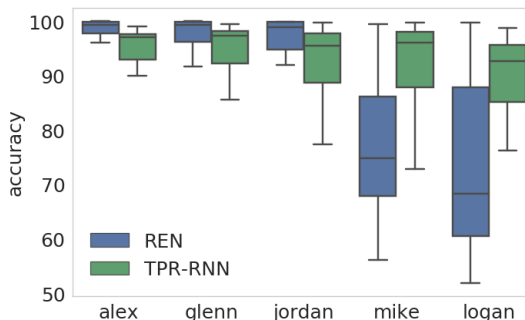

Figure 5: Average accuracy over the generated test sets of each task. The novel entities that we add to the training data were not trained on all tasks. For a model that generalises systematically, the test accuracy should not drop for entities with only partial training data.

stories the new entity is also contained in the question. We then add generated stories from all possible 40 combinations of new entities and tasks to the test set. To the training set, however, we only add stories from a subset of all tasks.

More specifically, the new entities are *Alex*, *Glenn*, *Jordan*, *Mike*, and *Logan* for which we generate training set stories from $8/8, 6/8, 4/8, 2/8, 1/8$ of the tasks respectively. We summarize the results in Figure 5 by averaging over tasks. After the network has been trained, we find that our model achieves high accuracy on entity/task pairs on which it has not been trained. This indicates its systematic generalisation capability due to the disentanglement of entities and relations.

Our analysis and the additional experiment indicate that the model seems to learn combinatorial representations resulting in interpretable distributed representations and data efficiency due to rule-like generalisation.

# 8   Limitations

To compute the correct gradients, an RNN with external memory trained by backpropagation through time must store all values of all temporary variables at every time step of a sequence. Since outer product-based Fast Weights [26, 27] and our TPR system have many more time-varying variables per learnable parameter than a classic RNN such as LSTM, this makes them less scalable in terms of memory requirements. The problem can be overcome through RTRL [2, 3], but only at the expense of greater time complexity. Nevertheless, our results illustrate how the advantages of TPRs can outweigh such disadvantages for problems of combinatorial nature.

One difficulty of our Fast Weight-like memory is the well-known vanishing gradient problem [56]. Due to multiplicative interaction of Fast Weights with RNN activations, forward and backward propagation is unstable and can result in vanishing or exploding activations and error signals. A similar effect may affect the forward pass if the values of the activations are not bounded by some activation function. Nevertheless, in our experiments, we abandoned bounded TPR values as they significantly slowed down learning with little benefit. Although our current sub-optimal initialization may occasionally lead to exploding activations and NaN values after the first few iterations of gradient descent, we did not observe any extreme cases after a few dozen successful steps, and therefore simply reinitialize the model in such cases.

A direct comparison with the DNC is a bit inconclusive for the following reasons. Our architecture, uses a sentence encoding layer similar to how many memory networks encode their input. This slightly facilitates the problem since the network doesn't have to learn which words belong to the same sentence. Most memory networks also iterate over sentence representations, which is less general than iterating over the word level, which is what the DNC does, which is even less general than iterating over the character level. In preliminary experiments, a word level variation of our architecture solved many tasks, but it may require non-trivial changes to solve all of them.

# 9   Conclusion

Our novel RNN-TPR combination learns to decompose natural language sentences into combinatorial components useful for reasoning. It outperforms previous models on the bAbI tasks through attentional control of memory. Our approach is related to Fast Weight architectures, another way of increasing the memory capacity of RNNs. An analysis of a trained model suggests straightforward interpretability of the learned representations. Our model generalises better than a previous state-of-the-art model when there are strong systematic differences between training and test data.

**Acknowledgments**

We thank Paulo Rauber, Klaus Greff, and Filipe Mutz for helpful comments and helping hands. We are also grateful to NVIDIA Corporation for donating a DGX-1 as part of the Pioneers of AI Research Award and to IBM for donating a Minsky machine. This research was supported by an European Research Council Advanced Grant (no: 742870).

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
