[Supplementary Material · Learning_to_Reason_with_Third_Order_Tensor_Products_v2_supplementary.pdf]

# Appendix

## A   Experimental Details

We encode the valid words for a task as a one-hot vector; the dimensionality of the vector space $V_{\text{Symbol}}$ is equal to the size of the vocabulary. Each MLP which produces the entity and relation representations from a sentence representation consists of two layers, where each layer is an affine transformation followed by the hyperbolic tangent nonlinearity. The hidden layers of the MLPs refer to the intermediate activations and are vectors from the vector space $V_{\text{Hidden}}$.

We initialize the word embeddings with a uniform distribution from $-0.01$ to $0.01$ and apply the Glorot initialization scheme [57] for all other weights except the position vectors which are initialized as a vector of ones divided by $k$, the number of position vectors. We implemented the model in the TensorFlow framework and compute the gradients through its automatic differentiation engine [58] based on Linnainmaa's automatic differentiation or backpropgation scheme [19]. We pad shorter sentences with the padding symbol to achieve a uniform sentence length but keep the story length dynamic as in previous work.

To deal with possible unstable initializations we incorporate a warm-up phase in which we train the network for 50 steps with $1/10$ of the learning rate. In the case of NaN values during this warm-up phase we reinitialize the network from scratch. After successful warm-up phases we never encountered any further instabilities.

We optimize the neural networks using the Nadam optimizer [59] which in our experiments consistently outperformed others in convergence speed but not necessarily in final performance. Finally, we multiply the learning rate by a factor of $0.5$ once it has reached a validation set loss below $0.1$.

**The Single-Task Model**   For the single-task model $\dim(V_{\text{Symbol}}) = \dim(V_{\text{Hidden}})$, $\dim(V_{\text{Entity}}) = 15$, and $\dim(V_{\text{Relation}}) = 10$. Note that $\dim(V_{\text{Symbol}})$ depends on the vocabulary size of each individual task. We achieved the results in Table 2 using the hyper-parameters learning-rate $= 0.008$, $\beta_1 = 0.6$, $\beta_2 = 0.4$, and a batch-size of 128. These hyper-parameters have been optimised using an evolution procedure with small random perturbations. The main effect is improved convergence speed. With the exception of a few tasks that were sensible to the momentum parameter, similar final performance can be achieved with the default hyper-parameters.

**The All-Tasks Model**   In the all-tasks setting we train one model on all tasks simultaneously. We increase the size of the model to $dim(V_{Hidden}) = 90$, $dim(V_{Entity}) = 40$, and $dim(V_{Relation}) = 20$ and train with a batch-size of 32. We used the default hyper-parameters learning-rate $= 0.001$, $\beta_1 = 0.9$, $\beta_2 = 0.999$ in that case.

# B    Detailed All-Tasks Training Runs

Table 4: Error percentage of 8 different TPR-RNNs in the all-tasks setting. The performance is further broken down into task-specific error percentages. We compare our all-tasks results to the previous state-of-the-art in Table 1.

| task | run-0 | run-1 | run-2 | run-3 | run-4 | run-5 | run-6 | run-7 | best | mean |
|------|-------|-------|-------|-------|-------|-------|-------|-------|------|------|
| all (0) | 1.50 | 1.69 | 1.13 | 1.04 | 0.78 | 0.96 | 1.20 | 2.40 | 0.78 | $1.34 \pm 0.52$ |
| 1 | 0.10 | 0.00 | 0.10 | 0.20 | 0.00 | 0.00 | 0.00 | 0.00 | 0.00 | $0.05 \pm 0.08$ |
| 2 | 1.70 | 0.80 | 0.60 | 0.30 | 0.40 | 0.50 | 0.50 | 0.30 | 0.30 | $0.64 \pm 0.46$ |
| 3 | 4.70 | 2.50 | 3.50 | 2.20 | 3.40 | 5.40 | 3.50 | 7.90 | 2.20 | $4.14 \pm 1.85$ |
| 4 | 0.00 | 0.00 | 0.00 | 0.10 | 0.20 | 0.10 | 0.00 | 0.00 | 0.00 | $0.05 \pm 0.08$ |
| 5 | 1.10 | 1.50 | 0.80 | 0.70 | 1.00 | 1.00 | 0.80 | 1.10 | 0.70 | $1.00 \pm 0.25$ |
| 6 | 0.00 | 1.10 | 0.70 | 0.10 | 0.10 | 0.40 | 0.00 | 0.50 | 0.00 | $0.36 \pm 0.39$ |
| 7 | 1.70 | 3.50 | 1.10 | 2.60 | 1.00 | 1.90 | 1.60 | 1.60 | 1.00 | $1.88 \pm 0.82$ |
| 8 | 0.20 | 1.40 | 0.40 | 0.40 | 0.50 | 0.40 | 0.30 | 0.50 | 0.20 | $0.51 \pm 0.37$ |
| 9 | 0.20 | 1.30 | 0.20 | 0.10 | 0.30 | 0.80 | 0.20 | 0.10 | 0.10 | $0.40 \pm 0.43$ |
| 10 | 1.40 | 2.40 | 1.20 | 0.30 | 0.40 | 0.20 | 0.40 | 0.80 | 0.20 | $0.89 \pm 0.75$ |
| 11 | 1.60 | 2.00 | 1.10 | 0.70 | 1.30 | 1.00 | 0.50 | 1.20 | 0.50 | $1.18 \pm 0.48$ |
| 12 | 1.30 | 1.00 | 2.60 | 1.00 | 0.20 | 0.00 | 3.40 | 1.30 | 0.00 | $1.35 \pm 1.14$ |
| 13 | 2.50 | 2.10 | 2.10 | 1.90 | 2.10 | 2.50 | 2.40 | 3.40 | 1.90 | $2.38 \pm 0.47$ |
| 14 | 0.80 | 0.20 | 0.70 | 1.90 | 0.20 | 0.90 | 1.00 | 1.10 | 0.20 | $0.85 \pm 0.54$ |
| 15 | 0.20 | 0.00 | 0.00 | 0.00 | 0.00 | 0.00 | 0.00 | 0.00 | 0.00 | $0.03 \pm 0.07$ |
| 16 | 0.20 | 0.20 | 0.10 | 4.00 | 0.40 | 0.00 | 0.60 | 0.10 | 0.00 | $0.70 \pm 1.35$ |
| 17 | 1.60 | 9.00 | 4.20 | 0.80 | 0.60 | 1.40 | 2.60 | 7.30 | 0.60 | $3.44 \pm 3.16$ |
| 18 | 0.20 | 1.60 | 1.30 | 0.70 | 0.00 | 0.70 | 1.20 | 0.10 | 0.00 | $0.72 \pm 0.60$ |
| 19 | 11.00 | 3.90 | 2.50 | 1.20 | 4.20 | 4.10 | 6.00 | 22.80 | 1.20 | $6.96 \pm 7.03$ |
| 20 | 0.00 | 0.00 | 0.00 | 0.00 | 0.00 | 0.00 | 0.00 | 0.00 | 0.00 | $0.00 \pm 0.00$ |

# C   Further Similarity Matrices

Figure 6: Hierarchically clustered sentences based on the cosine-similarity of the $e^{(1)}$ representation. We select sentences randomly from the test set and use our best all-tasks model to extract the representations. The sentences below are are also clustered. Note how they cluster together based on entity information like milk, green, or bathroom.

36: mary put down the milk there .
39: mary discarded the milk .
42: mary grabbed the milk there .
41: mary picked up the milk there .
45: jason picked up the milk there .
5: brian is green .
7: lily is green .
35: mary travelled to the kitchen .
43: jason went to the kitchen .
11: daniel went back to the kitchen .
17: sandra went back to the kitchen .
19: sandra travelled to the kitchen .
21: sandra journeyed to the kitchen .
33: mary went to the bathroom .
18: sandra went back to the bathroom .
49: john travelled to the bathroom .
14: daniel travelled to the bathroom .
12: daniel went to the bathroom .
15: daniel journeyed to the bathroom .
31: mary went back to the bedroom .
30: mary moved to the bedroom .
38: mary journeyed to the bedroom .
48: john went to the bedroom .
27: yann travelled to the bedroom .
10: daniel moved to the bedroom .

13: daniel travelled to the bedroom .
2: julius is white .
28: yann is tired .
8: sumit moved to the garden .
9: sumit is bored .
44: jason is thirsty .
34: mary travelled to the hallway .
26: bernhard is gray .
47: john went to the office .
20: sandra journeyed to the office .
32: mary went to the office .
3: julius is a rhino .
6: lily is a rhino .
22: sandra discarded the football .
23: sandra picked up the football there .
24: sandra took the football there .
46: then she journeyed to the bathroom .
1: greg is a frog .
4: brian is a frog .
29: yann picked up the pajamas there .
25: bernhard is a swan .
40: mary discarded the apple there .
16: daniel took the apple there .
37: mary got the apple there .

Figure 7: Hierarchically clustered sentences based on the cosine-similarity of the $e^{(2)}$ representation. We select sentences randomly from the test set and use our best all-tasks model to extract the representation. The sentences below are are also clustered. Note how the sentences cluster together based on entity information like milk, green, or bathroom.

30: mary went back to the kitchen .
28: mary moved to the hallway .
29: mary moved to the garden .
27: sandra journeyed to the bathroom .
24: sandra travelled to the bedroom .
22: sandra moved to the kitchen .
23: sandra went to the office .
3: daniel went back to the bathroom .
4: daniel went to the bathroom .
2: daniel moved to the kitchen .
1: daniel moved to the office .
5: daniel travelled to the office .
36: john travelled to the bathroom .
35: john went to the kitchen .
33: john moved to the office .
34: john went back to the bedroom .
6: daniel and john went to the hallway .
31: mary and sandra travelled to the hallway .
26: sandra left the apple .
37: john discarded the football .

19: the bathroom is south of the hallway .
20: the bathroom is north of the kitchen .
21: the bathroom is east of the garden .
12: the hallway is west of the office .
10: the hallway is north of the bedroom .
11: the hallway is east of the bedroom .
7: the office is east of the bathroom .
8: the office is west of the hallway .
9: the office is west of the garden .
38: john grabbed the football there .
32: mary picked up the milk there .
13: the garden is west of the bedroom .
25: sandra got the apple there .
17: the kitchen is south of the office .
18: the kitchen is south of the bathroom .
16: the bedroom is west of the garden .
14: the bedroom is east of the hallway .
15: the bedroom is east of the kitchen .

Figure 8: Hierarchically clustered questions based on the cosine-similarity of the $r^{(1)}$ representation. We select questions randomly from the test set and use our best all-tasks model to extract the representation. The questions below are are also clustered. Note how the sentences cluster together based on relation information like carrying, color, or afraid of.

4: is the pink rectangle to the left of the yellow square ?
3: is the red sphere below the triangle ?
1: is bill in the kitchen ?
0: does the chocolate fit in the container ?
5: is mary in the hallway ?
2: is julie in the park ?
6: is mary in the kitchen ?
14: how do you go from the bedroom to the office ?
13: how do you go from the garden to the hallway ?
21: what is john carrying ?
17: what is daniel carrying ?
19: what is mary carrying ?
15: how many objects is mary carrying ?
26: what did fred give to mary ?
22: what color is greg ?
24: what color is lily ?
23: what color is brian ?
25: what color is bernhard ?
12: where was fred before the office ?
10: where is mary ?
7: where is daniel ?
18: what is the kitchen south of ?
9: where is sandra ?
11: where is john ?
8: where is the milk ?
16: what is gertrude afraid of ?
20: what is emily afraid of ?