[Reviews · NeurIPS 2018]

Reviewer 1



Summary: This paper proposes to combine tensor product representations (TPRs) with RNNs for learning to reason. The entire model is end-to-end learnable including the TPRs. Experiments on bAbI tasks show that the proposed TPR-RNN is better than other state-of-the-art methods like RENs. Authors do some post analysis of learnt TPRs and claim that the results are human interpretable. TPR-RNN also shows systematic generalization to new entities. Comments: 1. How did you choose 15 entities and 10 relations in the experiments? What is the effect of choosing more entities or more relations? It would be good to have such an ablation study. 2. What is the intuition behind having 3 inference steps? Again it is good to have a study on what is the effect of increasing or decreasing the number of inference steps. 3. I appreciate authors’ effort to report the mean scores (with standard deviation) in bAbI task. Scores from the best runs are not very indicative of the performance of the algorithm. Mean scores give a much better picture. 4. Will you release the source code of the model to reproduce the results reported in the paper? I do not have any major concerns about this paper. The model is novel and the experimental results are convincing. There is enough technical novelty in the paper to accept it. Minor comments: 1. Line 27: line repeated. 2. Line 223: correct the sentence ending.

Reviewer 2



Summary This paper presents a question-answering system based on tensor product representations. Given a latent sentence encoding, different MLPs extract entity and relation representations which are then used to update an tensor product representations of order-3 and trained end-to-end from the downstream success of correctly answering the question. Experiments are limited to bAbI question answering, which is disappointing as this is a synthetic corpus with a simple known underlying triples structure. While the proposed system outperforms baselines like recurrent entity networks (RENs) by a small difference in mean error, RENs have also been applied to more real-world tasks such as the Children's Book Test (CBT). Strengths - I like that the authors do not just report the best performance of their model, but also the mean and variance from five runs. - I liked the ablation study, showing which operations are needed for which bAbI tasks. Weaknesses - Results on bAbI should be taken with a huge grain of salt and only serve as a unit-test. Specifically, since the bAbI corpus is generated from a simple grammar and sentence follow a strict triplet structure, it is not surprising to me that a model extracting three distinct symbol representations from a learned sentence representation (therefore reverse engineering the underlying symbolic nature of the data) would solve bAbI tasks. However, it is highly doubtful this method would perform well on actual natural language sentences. Hence, statements such as "trained [...] on a variety of natural language tasks" is misleading. The authors of the baseline model "recurrent entity networks" [12] have not stopped at bAbI, but also validated their models on more real-world data such as the Children's Book Test (CBT). Given that RENs solve all bAbI tasks and N2Ns solve all but one, it is not clear to me what the proposed method adds to a table other than a small reduction in mean error. Moreover, the N2N baseline in Table 2 is not introduced or referenced in this paper, so I am not sure which system the authors are referring to here. Minor Comments - L11: LSTMs have only achieved on some NLP tasks, whereas traditional methods still prevail on others, so stating they have achieved SotA in NLP is a bit too vague. - L15: Again, too vague, certain RNNs work well for certain natural language reasoning tasks. See for instance the literature on natural language inference and the leaderboard at https://nlp.stanford.edu/projects/snli/ - L16-18: The reinforcement learning / agent analogy seems a bit out-of-place here. I think you generally point to generalization capabilities which I believe are better illustrated by the examples you give later in the paper (from lines 229 to 253). - Eq. 1: This seems like a very specific choice of combining the information from entity representations and their types. Why is this a good choice? Why not keep the concatenation of the kitty/cat outer product and the mary/person outer product? Why is instead the superposition of all bindings a good design choice? - I believe section four could benefit from a small overview figures illustrating the computation graph that is constructed by the method. - Eq. 7: At first, I found it surprising why three distinct relation representation are extracted from the sentence representation, but it became clearer later with the write, move and backling functions. Maybe already mention at this point why the three relation representations are going to be used for. - Eq. 15: s_question has not been introduced before. I imagine it is a sentence encoding of the question and calculated similarly to Eq. 5? - Eq. 20: A bit more details for readers unfamiliar with bAbI or question answering would be good here. "valid words" here means possible answer words for the given story and question, correct? - L192: "glorot initalization" -> "Glorot initialization". Also, there is a reference for that method: Glorot, X., & Bengio, Y. (2010, March). Understanding the difficulty of training deep feedforward neural networks. In Proceedings of the thirteenth international conference on artificial intelligence and statistics (pp. 249-256). - L195: α=0.008, β₁=0.6 and β₂=0.4 look like rather arbitrary choices. Where does the configuration for these hyper-parameters come from? Did you perform a grid search? - L236-244: If I understand it correctly, at test time stories with new entities (Alex etc.) are generated. How does your model support a growing set of vocabulary words given that MLPs have parameters dependent on the vocabulary size (L188-191) and are fixed at test time? - L265: If exploding gradients are a problem, why don't you perform gradient clipping with a high value for the gradient norm to avoid NaNs appearing? Simply reinitializing the model is quite hacky. - p.9: Recurrent entity networks (RENs) [12] is not just an arXiv paper but has been published at ICLR 2017.

Reviewer 3



The authors propose three-dimensional Tensor Product Representations combined with RNNs for natural language reasoning tasks. The architecture is evaluated on the bAbI task, and it outperforms DNC. Pros: 1) The proposed architecture achieves state-of-the-art results on one natural language reasoning task with a rather simple structure. 2) The authors provide ablation study on most important components of the model. 3) In Section 7 they provide detailed analysis of the representations, to show that their reasoning model is interpretable. Cons: 1) Notation in Section 2 and 3 are confusing. There are \mathbf{f, r, a, b, u} in Section 2, and \mathbf{f, r, t, l, u} in Section 3, to represent entities. It is really hard to capture the basic idea from first reading. Please make the notations as succinct as possible. 2) As I understand, this architecture is inspired by Paul Smolensky's tensor memory model. My question is that why the sentence have to be mapped to a representation as in Eq.5; then representations of entities and relations are obtained from it via MLP? Why not just segment the sentence into entities and relations as motivated in Section 3, and combine them via tensor product? 3) My other concern is whether this model can be applied to more complicated NL reasoning tasks, since the authors compare their architecture with DNC.